# Small Renal Mass Cryoablation: Trifecta Outcomes of a Single-Institution Experience with a 20-Year Follow-Up

**DOI:** 10.3390/cancers17182960

**Published:** 2025-09-10

**Authors:** Mahdi Mottaghi, Alireza Ghoreifi, Sriram Deivasigamani, Sudharshanan Balaji, Eric S. Adams, Matvey Tsivian, Charles Y. Kim, Thomas J. Polascik

**Affiliations:** 1Department of Urology, Duke University Medical Center, Durham, NC 27710, USA; 2Section of Urology, Department of Surgery, Durham Veterans Affairs Health System, Durham, NC 27710, USA; 3Department of Urology, Hollings Cancer Center, Medical University of South Carolina, Charleston, SC 29425, USA; 4Department of Radiology, Duke University Medical Center, Durham, NC 27710, USA

**Keywords:** cryoablation, end stage renal disease, image-guided surgery, long term adverse effects, minimally invasive surgical procedures, renal carcinoma

## Abstract

Small kidney tumors (4 cm or less) could be managed with partial kidney removal, active monitoring, thermal ablation, or, less frequently, complete kidney removal, depending on the tumor characteristics and patient preference. Thermal ablation offers a minimally invasive approach for treating small kidney tumors and could be performed using heat (radiofrequency ablation) or cold (cryoablation). This research aimed to evaluate the long-term effectiveness of cryoablation in achieving (1) tumor control and (2) preserving kidney function, specifically in these patients. Our findings indicate that tumor control outcomes are largely achievable in patients with small kidney masses attributed to Renal Cell Carcinoma. Excluding those patients who had only one kidney, renal function was acceptable among the patients with a small kidney mass.

## 1. Introduction

The Surveillance, Epidemiology, and End Results (SEER) Program estimates that approximately 80,980 individuals will be diagnosed with kidney and renal pelvis cancer in 2025, ranking it as the seventh most common cancer type and accounting for 4.0% of all new cancer diagnoses in the United States [1,2]. Since the 1990s, the growing utilization of cross-sectional diagnostic imaging has led to a substantial rise in the incidental detection of renal masses, resulting in an approximately 1.5-fold increase in incidence [1]. Consequently, the related death rates declined from 4.3% in 1992 to 3.4% in 2023 [1]. It has been hypothesized that despite the rising incidence of renal cancer, its declining mortality in developed countries is largely attributed to the timely diagnosis of the Small Renal Masses (SRMs), secondary to increased diagnostic abdominal imaging utilization [3].

Approximately 70–80% of SRMs are malignant, though most exhibit indolent behavior and pose insignificant metastatic risk at the time of diagnosis [4]. While active surveillance is acknowledged as a management option in major urology and oncology guidelines, particularly for frail patients with limited life expectancy, nephron-sparing approaches have remained the preferred treatment choice for SRMs over the past two decades, given their advantages in preserving renal function while maintaining favorable oncologic outcomes [5,6,7]. Partial Nephrectomy (PN) stands as the gold-standard treatment for clinical stage T1 renal tumors. However, thermal ablative modalities, such as cryoablation, radiofrequency, and microwave ablation, are viable alternatives in appropriately selected patients, provided thorough counseling is conducted regarding the risks of tumor persistence or recurrence and the potential need for repeat ablation [8]. Despite the growing adoption of these novel treatments, which have demonstrated efficacy and low morbidity in the short- to intermediate-term, there remains a gap in the literature regarding long-term outcomes, specifically follow-ups beyond a decade [9]. This study aims to report long-term oncologic and renal functional outcomes following SRM cryoablation.

## 2. Materials and Methods

### 2.1. Study Design and Patient Population

We retrospectively reviewed a prospectively maintained renal mass database and included 158 consecutive patients who underwent 170 renal cryoablation procedures at our tertiary referral center between October 2001 and December 2011. The institutional review board approved the study. After excluding patients with syndromic etiologies of Renal Cell Carcinoma (RCC), such as Von Hippel–Lindau syndrome (3 cases), >4 cm tumor diameter (6 cases), and those with less than six months of follow-up (20 cases), 129 patients with an SRM were included in the analysis. All cases had preoperative cross-sectional contrast-enhanced imaging with Computed Tomography (CT) and/or Magnetic Resonance Imaging (MRI).

### 2.2. Surgical Intervention and Follow-Up

The decision to perform a renal biopsy before or during cryoablation was made based on physician judgment. Posterior lesions were preferentially managed with percutaneous cryoablation, while anteromedial lesions were favored for laparoscopic cryoablation due to their proximity to hilar structures. The choice between the two approaches was determined by physician discretion and patient preference. The percutaneous procedures were performed by the interventional radiology team, while the urology service conducted the laparoscopic interventions, as previously reported [10,11,12]. Follow-up assessments were scheduled at 3, 6, and 12 months post-procedure and annually thereafter. Serum creatinine levels were also measured at each follow-up visit, along with a general clinical assessment and evaluation for potential complications. Cross-sectional contrast-enhanced imaging was planned within two to six months after ablation to assess technical success, followed by surveillance at month twelve, and then annually for a minimum of three years post-ablation.

### 2.3. Study Variables and Outcomes

Data encompassing demographics (age, gender, race, body mass index), comorbidities (coronary arterial disease, diabetes, chronic obstructive pulmonary disease, dyslipidemia, hypertension, American Society of Anesthesiology score), smoking history, surgical history, imaging details (lesion size, RENAL nephrometry score), solitary kidney (SK) status, and radiologic and clinical follow-up variables were systematically collected.

The primary outcome included oncological efficacy, while secondary outcomes focused on functional efficacy and adverse events of SRM cryoablation. The oncological outcomes of interest, among those diagnosed with RCC, were local recurrence-free survival (RFS), metastasis-free survival (MFS), cancer-specific survival (CSS), and overall survival (OS). Procedural failure (persistence) was defined as the presence of enhancement on CT within the first three months after ablation and/or a positive biopsy, if applicable. If the first post-ablation imaging was indeterminate or postponed for any reason, the next imaging within the first six months from ablation was used to assess the technical failure. Local recurrence was defined as the appearance of a new enhancing or enlarging lesion within or at the ablated area after a technically successful ablation. Disease progression was categorized as local recurrence, metastasis, or need for systemic therapy. Complications were graded using the Clavien–Dindo system [13].

Functional outcomes were evaluated by measuring changes in creatinine levels and the estimated glomerular filtration rate (eGFR), which was calculated using the Modification of Diet in Renal Disease (MDRD) Study formula [14]. Regarding functional assessment time points, the relevant data were stratified into short-term (0–3 months), intermediate-term (4–12 months), and long-term (13–36 months) based on similar studies [15]. For patients on temporary dialysis within the first three months post-ablation, we only considered the creatinine measurement recorded immediately before dialysis initiation to account for acute kidney injury in short-term follow-up. Trifecta achievement was defined as the absence of disease progression at any point of follow-up, ≤10% decline in eGFR within 36 months of follow-up (<25% for SK patients), and the absence of grade 3–4 Clavien–Dindo complications [16]. The 10% and 25% eGFR cut-offs for non-SK and SK patients, respectively, were chosen based on prior literature [16,17]. The extended long-term (>36 months) renal functional outcomes were assessed by evaluating chronic kidney disease (CKD) stage progression, categorized as G1–G5 (≥90, 60–89, 30–59, 15–29, and <15 mL/min/1.73 m^2^). Additionally, we defined the term “expected eGFR” to be able to report even longer-term functional outcomes. In individuals without CKD and with normal kidney parenchyma, eGFR naturally declines from normal levels (≥90 mL/min/1.73 m^2^) by approximately 1 mL/min/1.73 m^2^ annually after the third decade of life [18,19,20]. Accordingly, “expected eGFR” was calculated based on the baseline (i.e., preoperative) eGFR, adjusted for an estimated decline of 0.5% per year.

### 2.4. Statistical Analysis

Continuous variables were presented as the median and Interquartile Range (IQR), while categorical variables were summarized as frequencies and percentages. The survival curves were estimated using Kaplan–Meier analysis. Creatinine and eGFR change over time were evaluated via the Generalized Linear Mixed Model, and Bonferroni post hoc adjustment was used for multiple pairwise comparisons. The threshold of statistical significance was *p* < 0.05. All statistical analyses were conducted using SPSS version 26 (IBM Corp., Somers, NY, USA) and Firth’s Cox Regression using R 4.2.0 (http://cran.r-project.org, accessed on 10 September 2024), and graphs were created using the ggplot2 package version 3.4.2.

## 3. Results

### 3.1. General Characteristics

A total of 129 patients (140 lesions) with a median age of 67 years (IQR 58–74) were included in the study. Among these, the median clinical and radiologic follow-up duration was 136 (IQR 54–180) and 74 (IQR 23–147) months, respectively. Biopsy data was available for 86 patients (67%). Among these, 62 (72%) were histologically diagnosed with RCC, and 24 (28%) exhibited benign pathologies, including angiomyolipoma in 3 patients, oncocytic neoplasm in 9 patients, and non-diagnostic pathology in 12 cases. Patient demographics are reported in Table 1.

Laparoscopic ablation was employed in 62 (48%), percutaneous ablation in 61 (48%), hand-assisted technique in 3 (2%), and robot-assisted laparoscopic technique in 3 (2%). Median preoperative creatinine and GFR were 1.0 mg/dL (IQR 0.8–1.3) and 68 mL/min/1.73 m^2^ (IQR 53–87), respectively. Table 2 shows tumor characteristics along with procedure-related information. Complete response to ablation (no enhancement on follow-up contrast CT) was achieved in 119/126 (94%) cases, while 7/126 (6%) of cases experienced radiologic procedure failure, and post-ablation images in three cases were not available within the first year of ablation for evaluating procedural success.

### 3.2. Adverse Events

Most cryoablation procedures (100/129; 78%) were conducted without adverse events. Of 29 reported complications, 24 were Clavien–Dindo grade 1–2, while 6 had higher-grade adverse events. The complication rate for any Clavien–Dindo grade was twice as high in patients with left-sided tumors (31%, 19/61) compared to those with right-sided tumors (15%, 10/68), showing a statistically significant difference (*p* = 0.026). Considering only cases with Clavien–Dindo grade 3–4 complications, four of those were left-sided and one right-sided, but this difference did not reach statistical significance (*p* = 0.191). Appendix A details these complications, along with their related management.

### 3.3. Functional Outcomes

Solitary Kidney (SK) and non-SK cases were analyzed separately, once at the 36-month follow-up and once among those with follow-ups exceeding 36 months.

#### 3.3.1. Non-SK Patients; 36-Month Follow-Up

For non-SK cases, the generalized mixed linear model with pairwise comparisons between time points showed a significant creatinine increase between baseline and 0–3 month intervals [mean difference (SE): 0.091 (0.021); *p* < 0.001]. Although statistically significant, the mean increment was ≤10% compared to the baseline. Of note, creatinine measures increased gradually from the first post-ablation checkpoint compared to later checkpoints, but this increment did not reach a statistically significant level (Table 3). A similar analysis revealed a statistically significant decline in eGFR measures post-ablation [mean difference (SE): −4.5 (1.46); *p* = 0.021] and in the third (12–24 month) checkpoint [mean difference (SE): −7.1 (2.30); *p* = 0.021] compared to pre-ablation (Table 3). Although statistically significant, the mean decline was ≤10% compared to the baseline. Of 116 non-SK patients, 81 had available creatinine between 13 and 36 months; 52 (64%) had a ≤10% creatinine elevation, and 51 (63%) had a ≤10% eGFR decline compared to baseline.

#### 3.3.2. SK Patients; 36-Month Follow-Up

Among 13 SK patients, renal function success, defined as <25% eGFR decline, was achieved in 54% (7/13). Of these, 29% (2/7) of biopsy-proven RCC patients and 83% (5/6) of the remainder of the SK cases were within the first 36-month follow-up.

#### 3.3.3. Non-SK Patients; >36-Month Follow-Up

We also evaluated the extended long-term (>36 months) renal function outcomes. Ninety-three cases had available creatinine measurements beyond 36 months; eight were SK cases at the time of ablation and thus were excluded from the long-term analysis. The remaining 85 non-SK patients had a mean creatinine follow-up of 138 ± 58 months (median 155, range 37–253) with a mean creatinine and eGFR of 1.45 ± 1.12 (median 1.2) mg/dL and 59 ± 26 (median 59), respectively. Mean creatinine increase and eGFR decline compared to baseline were 0.38 ± 0.97 (median 0.20) and 14 ± 22 (median 13) at their last follow-up. Two patients progressed to dialysis after 49 and 144 months. Of note other two patients underwent unilateral Radical Nephrectomy (RN) after 53 (Case #3, cause: local recurrence) and 104 (Case #99, left renal artery stenosis on the contralateral side to cryoablation) months following ablation, and their last creatinine before RN was used to assess long-term functional outcomes. One-stage and two-stage CKD progression were detected in 34/85 (40%) and 7/85 (8%) patients, respectively. Based on the definition of “expected eGFR,” 40/85 cases (47%) experienced ≤10% eGFR decline.

#### 3.3.4. SK Patients; >36-Month Follow-Up

Among non-SK patients, 9/106 (8%) eventually progressed to dialysis/transplantation after a median of 27 (IQR 7.5–78.5; range 6–144) months following ablation, with their median pre-ablation eGFR of 54 (IQR 36–102), while 2/13 (15%) of SK patients progressed to dialysis at 5 and 56 months post-ablation, with their pre-ablation eGFR levels of 36 and 30 mL/min/1.73 m^2^.

### 3.4. Oncological Outcomes (Only Biopsy-Proven RCC Cases)

Among patients with biopsy-proven RCC, the median overall survival time was 160 months (Figure 1). Within 5-, 10-, and 15-year follow-up timepoints, the OS rates were 81%, 60%, and 46%, and the MFS rates were 98%, 98%, and 91%, respectively. Local RFS was 91% at both 5- and 15-year follow-ups, as no local recurrences were identified after five years of follow-up. Of note, all three cases that developed metastasis had also been diagnosed with local recurrence simultaneously (in one case) or previously (in two cases). CSS rates were 98%, 96%, and 96% at 5, 10, and 15 years of follow-up.

Table 4 shows Firth’s Cox regression analysis results, which identified tumor volume (calculated by length × width^2^ × 0.52) and SK status at ablation as significant predictors for local recurrence. The same trend was observed in Firth’s multivariable Cox regression, although it did not reach statistical significance.

### 3.5. Trifecta Achievement

Of the 62 biopsy-proven RCC cases, 55 had data available for trifecta assessment. Considering seven SK cases, 58% (26/48) of the mentioned cohort achieved our trifecta definition at 36 months. Regarding SK patients, five RCC-proven cases experienced ≥25% eGFR decline; thus, only two of seven RCC-proven SK patients achieved the trifecta at the 36-month follow-up.

### 3.6. Follow-Up

Among all patients, 10 developed radiologic recurrence in the ablated part of the treated kidney; five belonged to the biopsy-proven RCC cohort, and five patients had no biopsy histology at cryoablation (Appendix B). Of note, the latter five patients were not included in the trifecta assessment, as further pathology studies showed non-malignant histology in RN/PN specimens. From these ten patients, two with radiologic recurrences were not classified as local recurrences. One had bilateral enhancing lesions at diagnosis, managed with PN (right, pathology RCC) and ablation (left). He was found to have local enhancement in the right kidney 15 years after PN, which was managed by radiofrequency ablation. One woman with a history of right RN for RCC 13 years prior developed multiple SRMs in the left kidney, managed by cryoablation (1 lesion, near hilum) and RFA (2 other polar lesions); recurrence occurred in the superior pole, for which she underwent PN and another cryoablation. Five years after the last intervention, she developed further recurrence and metastasis to the pancreatic head, ultimately succumbing to metastatic RCC. This is not an unexpected outcome given that she developed multiple distinct renal masses over 20 years from the first diagnosis of RCC. Thus, the recurrences in the two mentioned cases were not considered cryoablation-related.

## 4. Discussion

This study represents one of the longest follow-up series of patients undergoing cryoablation for an SRM. The primary endpoint of the present study was to evaluate the long-term oncological and renal functional outcomes of SRM cryoablation. Procedural success was achieved in 94% of cases, with minimal complications reported, most of which were low-grade. In addition, most patients clinically exhibited ≤10% eGFR decline. Extended long-term follow-up (median 155 months) showed that less than half of the non-SK patients experienced at least one stage of CKD progression. Oncological outcomes for biopsy-proven RCC were favorable, with 91% MFS and RFS at 15-year follow-up. These results underscore the effectiveness and durability of targeted cryoablation as a treatment for select patients, balancing oncological control with renal preservation.

Renal function preservation is one of the primary goals of cryoablation compared to more extensive surgeries. Although statistically significant post-ablation eGFR decline was observed, 64% of our non-SK patients experienced ≤10% decrement. Woldu et al. also showed 8.9 ± 13.4% eGFR decline within the first three months of SRM cryoablation, correlated with 8.6 ± 8.3% loss in renal parenchyma volume [21]. Similarly, Beemster and colleagues reported eGFR decline of 7 (95% CI 5–9) mL/min/1.73 m^2^ following SRM cryoablation and found that baseline eGFR is the sole predictor of renal insufficiency (eGFR < 30 mL/min/1.73 m^2^), while tumor size and SK status did not reach the statistically significant threshold [22]. Similar findings have been observed in several studies comparing ablative modalities, including cryoablation, with PN for stage cT1 renal masses, emphasizing the effectiveness of SRM cryoablation as a viable approach for preserving renal function [9]. However, the comparative functional efficacy against PN was notably lower in the SK cohort [16,23].

Assessing renal function during long-term follow-up is challenging due to the complex interplay among renal function, age, and other medical comorbidities. We were unable to find similar studies that examined renal function beyond our follow-up period. Thus, we propose the term “expected eGFR,” based on previous studies estimating the annual loss of renal function (16–18), that could serve as a creatinine-based measure for evaluating renal function in extended follow-up. However, this definition requires validation; therefore, we report CKD-stage progression to remain consistent with commonly accepted standards in the literature [24].

Regarding oncological outcomes, we found no progression (local recurrence or metastasis) after the first five years of follow-up. The 5-year OS, CSS, local RFS, and MFS rates were 81%, 98%, 91%, and 98% in our study, respectively. These values are similar to those of other studies [25,26], and the slight differences in OS measures possibly stem from the longer follow-up period in our study, the inherent selection bias, and baseline patient characteristics between the two studies [9]. Of note, the modest OS is likely related to the older age and more comorbidities in this patient population selected to undergo cryoablation. Comparative studies between ablation and PN have also shown comparable MFS and local RFS in multiple meta-analyses [9,27]. Furthermore, we found that higher tumor volume and SK status at ablation were significantly correlated development of local recurrence. This may be partially explained by the clinician’s more cautious approach to create a robust freeze with more generous margins in SK patients to minimize the risk of further deterioration in renal function. The higher risk of progression in SK patients following thermal ablation has been noted in the literature [16]. These findings highlight the excellent long-term oncological efficacy of SRM cryoablation when patients are appropriately selected.

Gill et al. introduced the trifecta concept for Partial Nephrectomy [17], which was later tailored to assess renal ablation outcomes [28]. Although the proposed model considered an eGFR decline of ≤10% as success, other studies used different cut-offs based on their studied cohort that must be considered when interpreting the results [16,28,29,30]. We used the same eGFR cut-off for non-SK cases and achieved trifecta in 58% of RCC-proven cases. Likewise, Lucignani et al. employed the same trifecta definition and evaluated 72 SRM patients treated with cryoablation with a median follow-up of 21 months (8–39) and reported a trifecta achievement rate of 59.5% [29]. Pandolfo and colleagues evaluated the outcomes of three ablation modalities, including cryoablation, and reported a trifecta achievement rate of 58.8%. However, they considered the eGFR decline of <25% as a successful renal functional outcome for completely endophytic, primarily cT1-stage renal masses [30]. This reflects the premise that tumor location is expected to impact the degree of renal function loss. The same group applied a similar cut-off when assessing outcomes following ablation of a stage cT1 renal mass in SK patients [16]. While these definitions are appropriate within the context of the studied cohorts, the decision to perform ablation remains multifactorial, wherein the achievement of the trifecta may not be possible with any treatment modality. Concerning future directions, a more standardized and comprehensive approach—potentially leveraging artificial intelligence—is still needed to integrate all contributing factors in predicting kidney function loss. This would provide a more precise benchmark for postoperative assessment while improving preoperative patient counseling and establishing realistic expectations regarding renal function.

Our study was not devoid of limitations. Despite being derived from a prospectively generated database, this study’s retrospective nature, lack of direct comparison with a PN cohort, potential selection bias, and the temporal differences in practice (modern focal therapy and imaging compared to the study period technology) during the 10-year study period should be considered when interpreting the results. Also, biopsy confirmation was not performed for all cases. However, to our knowledge, this study provides the longest follow-up in terms of renal functional outcomes following SRM cryoablation. It should also be noted that patients often struggle to return for long-term surveillance, especially when they feel cancer-free and their medical focus shifts over time. Further comparative studies with equivalent follow-up durations are necessary to validate these findings.

## 5. Conclusions

In this long-term report of SRM cryoablation outcomes, most patients achieved durable oncological control with low complication rates. Considering our strict criteria for functional outcomes and extended follow-up, renal function preservation was satisfactory in all cohorts, with superior long-term renal function observed in patients without solitary kidney status at ablation. These findings highlight the need for careful patient selection as well as additional research with long-term follow-up in terms of renal functional outcomes, parallel to oncological outcomes.

## Figures and Tables

**Figure 1 cancers-17-02960-f001:**
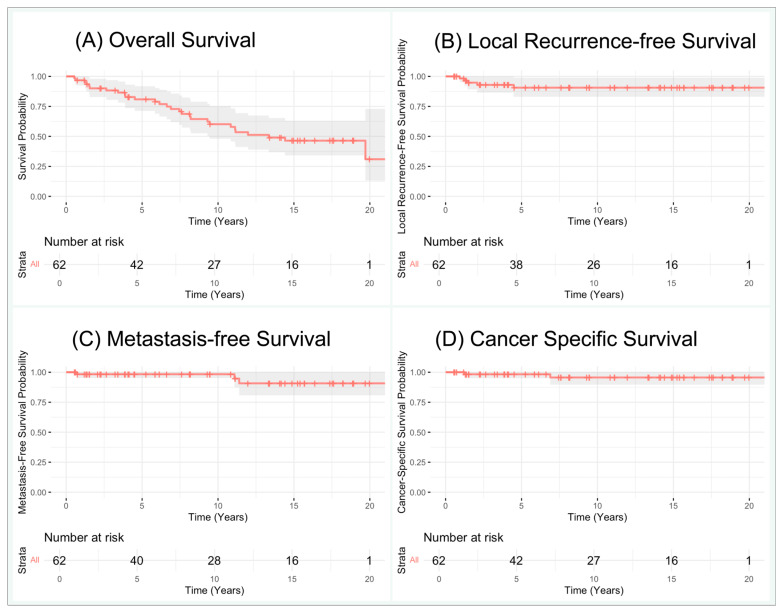
Kaplan–Meier curves for biopsy-proven Renal Cell Carcinomas: (**A**) overall survival, (**B**) local recurrence-free survival, (**C**) metastasis-free survival, (**D**) cancer-specific survival.

**Table 1 cancers-17-02960-t001:** Patient demographics. Abbreviations: RCC: Renal Cell Carcinoma; BMI: Body Mass Index; CAD: Coronary Arterial Disease; DM: Diabetes Mellitus; COPD: Chronic Obstructive Pulmonary Disease; IQR: Interquartile Range; PVD: Peripheral Vascular Disease.

Status	Overall (129, 100%)	RCC (62, 48%)	No Biopsy (43, 33%)	Benign Pathology (24, 19%)
Age (Years)				
Median (IQR)	67 (58–74)	67 (59–72)	68 (55–78)	68 (59–73)
Gender (%)				
Female	47 (36)	20 (32)	18 (42)	9 (37)
Male	82 (64)	42 (68)	25 (58)	15 (63)
Race (%)				
Black	42 (33)	18 (29)	18 (42)	6 (25)
White	83 (64)	43 (69)	22 (51)	18 (75)
Other	4 (3)	1 (2)	3 (7)	None
BMI				
Median (IQR)	28.1 (25–32)	28.7 (25.7–32.6)	27.70 (24.8–32.6)	27.9 (25.4–30.8)
Comorbidities (%)				
CAD (%)	23 (18)	15 (24)	4 (10)	4 (17)
DM (%)	35 (27)	16 (26)	12 (29)	7 (29)
Hypertension (%)	93 (72)	42 (68)	33 (79)	18 (75)
COPD (%)	11 (9)	6 (10)	4 (10)	1 (4)
Dyslipidemia (%)	47 (36)	21 (34)	14 (33)	12 (50)
PVD (%)	14 (11)	8 (13)	2 (5)	4 (17)
Solitary Kidney (%)	13 (10)	7 (11)	5 (12)	1 (4)
Renal Transplant (%)	2 (2)	0 (0)	2 (5)	0 (0)
Smoking (%)	63 (49)	33 (53)	17 (41)	13 (54)

**Table 2 cancers-17-02960-t002:** Tumor characteristics and details related to the cryoablation procedure. Abbreviations: IQR: Interquartile Range; RCC: Renal Cell Carcinoma.

Status	Overall (129 Patients)	RCC (62 Patients)	No Biopsy (43 Patients)	Benign Pathology (24 Patients)
Tumor Length (cm)				
Median (IQR)	2.0 (1.6–2.6)	2.2 (1.6–2.6)	2.1 (1.8–2.6)	1.7 (1.6–1.9)
Tumor Width (cm)				
Median (IQR)	1.7 (1.3–2.3)	1.8 (1.5–2.3)	1.7 (1.3–2.3)	1.6 (1.3–1.9)
Tumor volume * (cc)				
Median (IQR)	3.3 (1.6–6.6)	3.7 (1.9–6.9)	3.2 (1.6–6.6)	2.4 (1.2–3.8)
Number of Probes				
Median (IQR)	3 (2–4)	4 (3–4)	2 (2–3)	3 (2–4)
RENAL Nephrometry Scores				
Median (IQR)	6.0 (5–8)	6 (6–7)	7 (6–8)	6.5 (5–7)
Pole (%)				
Lower	45/127 (35)	19/61 (31)	18/44 (41)	8/22 (36)
Central	54/127 (43)	28/61 (46)	15/44 (34)	11/22 (50)
Upper	28/127 (22)	14/61 (23)	11/44 (25)	3/22 (14)
Location (%)				
Anterior	46/120 (38)	27/61 (38)	12/36 (33)	7/23 (31)
Posterior	40/120 (33)	17/61 (28)	12/36 (33)	11/23 (48)
Lateral	27/120 (23)	14/61 (23)	9/36 (25)	4/23 (17)
Medial	7/120 (6)	3/61 (5)	3/36 (8)	1/23 (4)
Hilar (%)	1/108 (1)	1/57 (2)	0/33 (0)	0/18 (0)
Laterality (%)				
Left	61/129 (47)	36/62 (58)	15/43 (35)	10/24 (42)
Right	68/129 (53)	26/62 (42)	28/43 (65)	14/24 (58)
Polar Line Crossing				
Entirely higher/Lower	31/108 (29)	15/57 (26)	10/33 (31)	6/18 (33)
Cross <50%	32/108 (30)	17/57 (30)	8/33 (23)	7/18 (39)
Cross ≥50%	11/108 (10)	7/57 (12)	3/33 (9)	1/18 (6)
Between lines	16/108 (15)	9/57 (16)	4/33 (11)	3/18 (16)
Cross axial midline	18/108 (17)	9/57 (16)	8/33 (26)	1/18 (6)
Distance from Collecting system				
≤4 mm	38/108 (35)	18/57 (31)	15/33 (45)	5/18 (28)
4–7 mm	23/108 (21)	14/57 (25)	3/33 (10)	6/18 (33)
≥7 mm	47/108 (43)	25/57 (44)	15/33 (45)	7/18 (39)
Duration of Hospitalization for Initial Cryoablation (days)				
Median (IQR)	1.0 (0.0–1.0)	1 (1.0–2.0)	0.0 (0.0–0.0)	1 (1.0–1.5)

* Tumor volume calculated as follows: length × (width^2^) × 0.52.

**Table 3 cancers-17-02960-t003:** Renal functional outcomes, excluding solitary kidney cases. Abbreviations: eGFR: estimated Glomerular Filtration Rate.

Time Point I	Time Point J	Mean Creatinine (SE) (mg/dL)	95% CI	Pairwise Comparisons	Mean Difference [I-J] (SE) **	*p*-Value *
Preoperative		1.15 (0.07)	1.01–1.28	reference	reference	reference
	0–3 months	1.24 (0.07)	1.10–1.38	vs. preoperative	0.09 (0.02)	<0.001
	3–12 months	1.26 (0.09)	1.08–1.44	vs. preoperative	0.11 (0.04)	0.047
	12–24 months	1.29 (0.09)	1.12–1.46	vs. preoperative	0.15 (0.04)	0.007
	24–36 months	1.31 (0.09)	1.14–1.48	vs. preoperative	0.16 (0.05)	0.023
	24–36 months	-	-	vs. 0–3 months	0.07 (0.05)	>0.99
**Time Point I**	**Time Point J**	**Mean eGFR (SE) (mL/min/1.73 m^2^)**	**95% CI**	**Pairwise Comparisons**	**Mean Difference [I-J] (SE) ****	** *p* ** **-Value**
Preoperative		71.9 (2.4)	67.3–76.6	reference	reference	reference
	0–3 months	67.4 (2.5)	62.5–72.3	vs. preoperative	−4.5 (1.5)	0.021
	3–12 months	67.6 (2.4)	62.8–72.3	vs. preoperative	−4.3 (1.8)	0.186
	12–24 months	64.8 (2.6)	59.6–69.9	vs. preoperative	−7.1 (2.3)	0.021
	24–36 months	65.8 (3.0)	58.8–70.6	vs. preoperative	−7.1 (2.9)	0.13
	24–36 months	-	-	vs. 0–3 months	−2.6 (2.8)	>0.99

* Pairwise comparisons performed using the Linear Mixed Model with Bonferroni. ** The “I” and “J” are arbitrary labels assigned by the software to differentiate the two contrasting values in pairwise tests and mentioned here for better illustration of the differences.

**Table 4 cancers-17-02960-t004:** Cox regression analysis for local recurrence-free survival. Firth’s Penalized Likelihood method was used due to the low number of events (five progressions as events). Abbreviations: ASA: American Society of Anesthesiology; BMI: Body Mass Index; COPD: Chronic Obstructive Pulmonary Disease; eGFR: estimated Glomerular Filtration Rate.

Variable	Unadjusted HR	*p*	*n*	Adjusted HR	*p*	*n*
Female Sex	0.68 (0.07, 3.65)	0.67	62			
Black Race	0.21 (0.00, 1.82)	0.18	62			
Age	1.04 (0.96, 1.14)	0.42	62			
BMI	0.98 (0.84, 1.13)	0.82	61			
Smoking	1.32 (0.26, 7.96)	0.73	62			
Hypertension	4.97 (0.56, 653.38)	0.18	62			
Diabetes	0.24 (0.00, 2.13)	0.24	62			
Dyslipidemia	4.55 (0.75–47.02)	0.1	62			
Coronary Arterial Disease	2.42 (0.40, 12.52)	0.31	62			
COPD	0.96 (0.01, 8.48)	0.98	62			
ASA Score	0.47 (0.04, 4.91)	0.52	40			
History of Past Abdominal Surgery	0.96 (0.19, 5.75)	0.96	62			
Distance from collecting system ≤7 mm	1.74 (0.29, 18.01)	0.56	57			
Distance from collecting system ≤4 mm	0.93 (0.09, 5.73)	0.94	57			
Exophytic Mass (Any)	0.83 (0.09, 110.83)	0.91	57			
Exophytic Mass >50%	0.54 (0.08, 3.50)	0.5	57			
Location to Polar Lines	1.81 (0.92, 4.46)	0.09	57			
Preoperative Creatinine	1.20 (0.06, 9.52)	0.88	61			
eGFR Pre-ablation	0.99 (0.95, 1.03)	0.57	61			
eGFR Post-ablation	0.98 (0.94, 1.02)	0.36	39			
Percutaneous vs. Laparoscopic	5.99 (0.79, 65.58)	0.08	60			
Number of Ablation Probes	1.13 (0.59, 2.11)	0.71	62			
Solitary Kidney at Ablation	6.69 (1.11, 34.56)	0.04	62	9.05 (0.52, 128)	0.11	51
Clavien–Dindo Grade (1–4)	0.77 (0.15, 1.75)	0.59	62			
Clavien–Dindo Grade > 2	1.25 (0.01, 11.08)	0.88	62			
Tumor Diameter	2.30 (0.84, 6.21)	0.1	62			
Tumor Volume (L × W^2^ × 0.52)	1.12 (1.02, 1.25)	0.03	51	1.1 (1.00, 1.22)	0.06	51

## Data Availability

Data used in the analysis of this study is available through the corresponding author upon reasonable request and adherence to relevant institutional regulations.

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
