# Peer review of "Small Renal Mass Cryoablation: Trifecta Outcomes of a Single-Institution Experience with a 20-Year Follow-Up"

_cancers, 2025, doi:10.3390/cancers17182960_

Round 1

Reviewer 1 Report

Comments and Suggestions for Authors

The manuscript presents an impressive long-term dataset on cryoablation outcomes for small renal masses, and the authors are commended for their comprehensive and structured analysis. However, the manuscript would benefit from some revisions to enhance clarity, scientific rigor, and manuscript polish.

First, while the introduction provides appropriate background, it should more clearly articulate the gap in current literature that this study fills, especially regarding the novelty of the 20-year follow-up.

Methodologically, the definitions of outcomes, particularly for "trifecta" and "expected eGFR," should be more explicitly justified with references or comparative discussion, and the rationale for thresholds (e.g., ≤10% eGFR decline) should be elaborated.

The results section is robust but could be more concise, with clearer separation between SK and non-SK cohorts and perhaps a summary table for key oncologic and renal outcomes across timepoints.

Statistical reporting is appropriate, but the manuscript should clarify whether any adjustments were made for multiple comparisons beyond Bonferroni in linear models.

In the discussion, while the study is well contextualized in the literature, it would be strengthened by more critically assessing the limitations of retrospective design, including potential selection bias, incomplete biopsy data, and variability in imaging protocols across the study period.

Lastly, the language throughout the manuscript, though generally sound, contains typographical and formatting inconsistencies (e.g., spacing, subscript/superscript, and special characters) that should be addressed to meet publication standards.

Overall, the study is of high potential impact and would be significantly improved with these revisions.

Author Response

Dear esteemed editorial team,    

I am pleased to submit the revised version of our manuscript titled "Small Renal Mass Cryoablation: Trifecta Outcomes of a Single-Institution Experience with 20-year Follow-up" to the Cancers Journal.

Thank you very much for taking the time to review this manuscript. Please find the detailed responses below and the corresponding revisions highlighted in the re-submitted files.

Thank you for the opportunity to publish our work in the Cancers journal.

Best regards,

Mahdi Mottaghi, M.D.

Reviewer 1 Comments

A) Questions for General Evaluation

Reviewer’s Evaluation

Response and Revisions

Does the introduction provide sufficient background and include all relevant references?

Yes

Improvements were made in yellow highlight.

Is the research design appropriate?

Yes

Are the methods adequately described?

Yes

Are the results clearly presented?

Can be improved

Are the conclusions supported by the results?

Can be improved

Are all figures and tables clear and well-presented?

Yes

Open Review

( ) I would not like to sign my review report

(x) I would like to sign my review report

Quality of English Language

( ) The English could be improved to more clearly express the research.

(x) The English is fine and does not require any improvement.

B) Point-by-point response to Comments

Comment 1: The manuscript presents an impressive long-term dataset on cryoablation outcomes for small renal masses, and the authors are commended for their comprehensive and structured analysis. However, the manuscript would benefit from some revisions to enhance clarity, scientific rigor, and manuscript polish.

First, while the introduction provides appropriate background, it should more clearly articulate the gap in current literature that this study fills, especially regarding the novelty of the 20-year follow-up.

Response: Thank you for your comment. This has been addressed as follows:

    Edited as page 2, Introduction, last paragraph, line 70: “Despite the growing adoption of these novel treatments, which have demonstrated efficacy and low morbidity in the short- to intermediate-term, there remains a gap in the literature regarding long-term outcomes, specifically follow-ups beyond a decade [9].”

Comment 2: Methodologically, the definitions of outcomes, particularly for "trifecta" and "expected eGFR," should be more explicitly justified with references or comparative discussion, and the rationale for thresholds (e.g., ≤10% eGFR decline) should be elaborated.

Response: Thank you for this relevant comment. We introduced the term “expected eGFR” to better report long-term kidney function outcomes. In individuals who do not have CKD, eGFR tends to decrease naturally over time. Specifically, after the age of 30, these individuals typically experience a decline of about 1 mL/min/1.73 m² per year. This decline starts in early adulthood and continues gradually with age. To quantify what might be considered a normal or expected decline, we calculated an "expected eGFR' for each person based on their pre-operative (before surgery) eGFR measurement. This calculation adjusts the baseline eGFR by accounting for an estimated decline of approximately 0.5% each year, reflecting the typical age-related decrease in kidney function. With the mentioned background in mind, we defined "expected eGFR" that was calculated based on the baseline (i.e., pre-operative) eGFR, adjusted for an estimated decline of 0.5% per year.

    Mentioned in page 4, Materials and Methods (Study variables and outcomes), line 129: “The extended long-term (>36 months) renal functional outcomes were assessed by evaluating chronic kidney disease (CKD) stage progression, categorized as G1-G5 (≥90, 60–89, 30–59, 15–29, and <15 mL/min/1.73 m²). Additionally, we defined the term “ex-pected eGFR” to be able to report even longer-term functional outcomes. In individuals without CKD and with normal kidney parenchyma, eGFR naturally declines from normal levels (≥90 mL/min/1.73 m²) by approximately 1 mL/min/1.73 m² annually after the third decade of life [17-19]. Accordingly, "expected eGFR" was calculated based on the baseline (i.e., pre-operative) eGFR, adjusted for an estimated decline of 0.5% per year.”

The eGFR decline cut-offs were chosen based on similar studies.

   Edited as page 4, Materials and Methods (Study variables and outcomes), line 128: “The 10% and 25% eGFR cut-offs for non-SK and SK patients, respectively, were chosen based on prior literature [16,17].”

Comment 3: The results section is robust but could be more concise, with clearer separation between SK and non-SK cohorts and perhaps a summary table for key oncologic and renal outcomes across timepoints. Statistical reporting is appropriate, but the manuscript should clarify whether any adjustments were made for multiple comparisons beyond Bonferroni in linear models.

Response: Thank you for this constructive feedback. We restructured the results, “section 3.3.” to:

3.3.1 Non-SK Patients; 36-month follow-up

3.3.2. SK Patients; 36-month follow-up

3.3.3. Non-SK Patients; >36-month follow-up

3.3.4. SK Patients; >36-month follow-up”.

Because we have reached the limit for number of tables in the manuscript, we added a graphical abstract instead, to better incorporate the reviewer’s point and stick to the journal’s standards.

We did only use Bonferroni post-hoc adjustment for pairwise comparisons in the generalized linear mixed model, as mentioned in the statistical analysis section, and we made this point more explicit to address your query.

   Edited as page 4, Materials and Methods (Statistical analysis section), line 140: “Creatinine and eGFR change over time were evaluated via the Generalized Linear Mixed Model and Bonferroni post-hoc adjustment was used for multiple pairwise comparisons.”

Comment 4: In the discussion, while the study is well contextualized in the literature, it would be strengthened by more critically assessing the limitations of retrospective design, including potential selection bias, incomplete biopsy data, and variability in imaging protocols across the study period.

Response: Thank you for your comment. Addressed as suggested:

     Edited as page 8, Discussion, last paragraph, line 331: ”Our study was not devoid of limitations. Despite being derived from a prospectively generated database, this study's retrospective nature, lack of direct comparison with a PN cohort, potential selection bias, and the temporal differences in practice (modern focal therapy and imaging compared to the study period technology) during the 10-year study period should be considered when interpreting the results. Also, biopsy confirmation was not performed for all cases.”

Comment 5: Lastly, the language throughout the manuscript, though generally sound, contains typographical and formatting inconsistencies (e.g., spacing, subscript/superscript, and special characters) that should be addressed to meet publication standards.

Overall, the study is of high potential impact and would be significantly improved with these revisions.

Response: Thank you for bringing this to our attention. We tried to address these issues to our best ability by asking authors who are native English speakers to re-read the text for punctuation correction, spacing, etc. The changes were made in yellow highlight.

Again, we want to appreciate the reviewer for the comments and helping us improve this draft.

Reviewer 2 Report

Comments and Suggestions for Authors

Title - clearly stating the essence of the study - No remarks

Abstract - concise review of the study - No remarks

Introduction - in-depth presentation of the contemporary literature, clearly defining the problem that this study aims to investigate - No remarks

Material and methods - comprehensive and highly sophisticated protocol - No remarks

Results - Trifecta achievement should be assessed in the group with confirmed RCC, but also in the group without pathology (5 cases of radiological recurrence were in the group without biopsy) - Major

Discussion and conclusions - the study has significant limitations, that authors has mentioned and taken into consideration, when interpreting the results - the major one is the lack of control group on active surveillance and comparator active group with PN; the differences in modern focal therapy compared to the study period is significant, but the time gap is inevitable in order to discuss long term results; lack of biopsy confirmation is a significant drawback and limits the firm credibility of the results to the RCC group (62 patients). Authors have acknowledge that and recommend to avoid this limitations in validation studies

Author Response

Dear esteemed editorial team,    

I am pleased to submit the revised version of our manuscript titled "Small Renal Mass Cryoablation: Trifecta Outcomes of a Single-Institution Experience with 20-year Follow-up" to the Cancers Journal.

Thank you very much for taking the time to review this manuscript. Please find the detailed responses below and the corresponding revisions highlighted in the re-submitted files.

Thank you for the opportunity to publish our work in the Cancers journal.

Best regards,

Mahdi Mottaghi, M.D.

Reviewer 2 Comments

A) Questions for General Evaluation

Reviewer’s Evaluation

Response and Revisions

Does the introduction provide sufficient background and include all relevant references?

Yes

Improvements were made in yellow highlight.

Is the research design appropriate?

Can be improved

Are the methods adequately described?

Can be improved

Are the results clearly presented?

Must be improved

Are the conclusions supported by the results?

Can be improved

Are all figures and tables clear and well-presented?

Yes

Open Review

( ) I would not like to sign my review report

(x) I would like to sign my review report

Quality of English Language

( ) The English could be improved to more clearly express the research.

(x) The English is fine and does not require any improvement.

B) Point-by-point response to Comments

Comment 1: Title - clearly stating the essence of the study - No remarks

Abstract - concise review of the study - No remarks

Introduction - in-depth presentation of the contemporary literature, clearly defining the problem that this study aims to investigate - No remarks

Material and methods - comprehensive and highly sophisticated protocol - No remarks

Response: We would like to thank the reviewer for their time, consideration, and kind comments.

Comment 2: Results - Trifecta achievement should be assessed in the group with confirmed RCC, but also in the group without pathology (5 cases of radiological recurrence were in the group without biopsy) - Major.

Response: Thank you for pointing out such an important point. This was one of the concerns between co-authors during internal team review of the manuscript and was one of the reasons we provided the table in Appendix 2. Three of those five cases had non-RCC in pathology reports of RN (1 oncocytoma, 1 normal kidney tissue, and 1 non-diagnostic for malignancy). Additionally, oncological control was defined as lack of metastasis/local recurrence while those non-malignant pathologies do not have the potential for metastasis and thus, have not put in the same group with biopsy-confirmed RCC cases. As a result, we decided not to include these in the trifecta assessment. However, we added this explanation to the text for a better clarification. Again, thank you for this important comment.

    Edited as page 8, Results (3.6. Follow-up paragraph), line 246: “Of note, the latter five patients were not included in the trifecta assessment as further pathology studies showed non-malignant histology in RN/PN specimens.”

Comment 3: Discussion and conclusions - the study has significant limitations, that authors has mentioned and taken into consideration, when interpreting the results - the major one is the lack of control group on active surveillance and comparator active group with PN; the differences in modern focal therapy compared to the study period is significant, but the time gap is inevitable in order to discuss long term results; lack of biopsy confirmation is a significant drawback and limits the firm credibility of the results to the RCC group (62 patients). Authors have acknowledge that and recommend to avoid this limitations in validation studies.

Response: Thank you for this comment. We totally agree with this comment and we made a change to the wording of the sentences to better address the reviewer’s comment:

     Edited as page 8, Discussion, last paragraph, line 331: ”Our study was not devoid of limitations. Despite being derived from a prospectively generated database, this study's retrospective nature, lack of direct comparison with a PN cohort, potential selection bias, and the temporal differences in practice (modern focal therapy and imaging compared to the study period technology) during the 10-year study period should be considered when interpreting the results. Also, biopsy confirmation was not performed for all cases.”

Reviewer 3 Report

Comments and Suggestions for Authors

This is an interesting single-institution report on small renal mass cryoablation with a remarkably long follow-up period. The topic is relevant, and the authors should be commended for their 20-year dataset. However, the manuscript presents significant methodological limitations that limit its current value to the literature.

Major comments

  1. Inclusion and exclusion criteria: These are not sufficiently detailed. Clear definitions are necessary to ensure reproducibility and allow readers to understand patient selection and potential biases.
  2. Lack of comparator group: The absence of a control cohort—such as a matched group undergoing RAPN, as in previously published series (e.g., https://pubmed.ncbi.nlm.nih.gov/36216659/)—significantly limits the ability to draw conclusions about the relative efficacy or safety of cryoablation. At the very least, a historical comparator would help contextualize outcomes.
  3. High proportion of benign lesions: The large number of benign histologies reduces the applicability of the current “trifecta” endpoint, which is largely designed for malignant cases. This could be an opportunity for the authors to propose a novel, perioperative-focused composite endpoint more suited to mixed malignant/benign populations.
  4. Statistical modelling concerns: The use of univariate and multivariate regression is questionable given the small number of events relative to the number of covariates. As a rule of thumb, at least 10 events per variable are required to avoid model overfitting. The current stepwise approach risks instability of estimates. I recommend limiting the model to clinically relevant and statistically significant univariate predictors, and clearly justifying their inclusion in the multivariate analysis while preserving all variables to provide a more robust model without overfitting.
  5. Novelty: Several studies with larger cohorts and direct comparisons to surgical approaches have already been published. The authors should explicitly state what new insights this dataset provides beyond confirming existing evidence. Without a comparator and with the noted methodological constraints, the novelty appears limited.

Author Response

Dear esteemed editorial team,    

I am pleased to submit the revised version of our manuscript titled "Small Renal Mass Cryoablation: Trifecta Outcomes of a Single-Institution Experience with 20-year Follow-up" to the Cancers Journal.

Thank you very much for taking the time to review this manuscript. Please find the detailed responses below and the corresponding revisions highlighted in the re-submitted files.

Thank you for the opportunity to publish our work in the Cancers journal.

Best regards,

Mahdi Mottaghi, M.D.

Reviewer 3 Comments

A) Questions for General Evaluation

Reviewer’s Evaluation

Response and Revisions

Does the introduction provide sufficient background and include all relevant references?

Yes

Improvements were made in yellow highlight.

Is the research design appropriate?

Must be improved

Are the methods adequately described?

Must be improved

Are the results clearly presented?

Yes

Are the conclusions supported by the results?

Can be improved

Are all figures and tables clear and well-presented?

Yes

Open Review

(x) I would not like to sign my review report

( ) I would like to sign my review report

Quality of English Language

( ) The English could be improved to more clearly express the research.

(x) The English is fine and does not require any improvement.

B) Point-by-point response to Comments

Comment 1: This is an interesting single-institution report on small renal mass cryoablation with a remarkably long follow-up period. The topic is relevant, and the authors should be commended for their 20-year dataset. However, the manuscript presents significant methodological limitations that limit its current value to the literature. Major comments:

Inclusion and exclusion criteria: These are not sufficiently detailed. Clear definitions are necessary to ensure reproducibility and allow readers to understand patient selection and potential biases

Response: Thank you for bringing this to our attention. Patients with sporadic SRM were included and those with syndromic etiologies of RCC, such as Von Hippel-Lindau syndrome were excluded. This clarified as below:

   Edited as page 2, Materials and Methods, paragraph 1: “We retrospectively reviewed a prospectively maintained renal mass database and included 158 consecutive patients who underwent 170 renal cryoablation procedures at our tertiary referral center between October 2001 and December 2011. The institutional review board approved the study. After excluding patients with syndromic etiologies of renal cell carcinoma (RCC), such as Von Hippel–Lindau syndrome (3 cases), >4cm tumor diameter (6 cases), and those with less than six months of follow-up (20 cases), 129 patients with a SRM were included in the analysis.”

Comment 2: Lack of comparator group: The absence of a control cohort—such as a matched group undergoing RAPN, as in previously published series (e.g., https://pubmed.ncbi.nlm.nih.gov/ 36216659/)—significantly limits the ability to draw conclusions about the relative efficacy or safety of cryoablation. At the very least, a historical comparator would help contextualize outcomes.

Response: Thank you for the comment. This is a very relevant comment. This has been the target of our project from the beginning, but extracting consecutive patients who underwent PN for SRM during the study interval is not possible currently for us. It may take 18 months to update such database and assign nephrometry scores and follow-ups by a dedicated radiologist (the median follow-up in the mentioned study was 43 months compared to 136 months in our study, making it more time-consuming to update similar PN cohort). Thus, we mentioned this as a limitation to our study: Page 11, discussion, last paragraph: “Our study was not devoid of limitations. Despite being derived from a prospectively generated database, this study's retrospective nature, lack of direct comparison with a PN cohort, and the temporal differences in practice during the 10-year study period should be considered when interpreting the results.”

Comment 3: High proportion of benign lesions: The large number of benign histologies reduces the applicability of the current “trifecta” endpoint, which is largely designed for malignant cases. This could be an opportunity for the authors to propose a novel, perioperative-focused composite endpoint more suited to mixed malignant/benign populations.

Response: Thank you for your comment. We agree that our study’s notable proportion of benign histologies makes applying a traditional endpoint challenging, as that metric is largely tailored to malignant outcomes. We have carefully considered your suggestion to propose a novel composite endpoint. However, we are concerned that combining these distinct patient groups could introduce a significant bias. Malignant cases necessitate more intensive, long-term follow-up for recurrence monitoring, which in turn facilitates a more complete assessment of long-term renal function. Patients with benign lesions, by contrast, do not have the same follow-up schedule. For these reasons, we believe that reporting perioperative complications and renal function separately for the two groups provides a clearer and more clinically relevant analysis. This approach, in our view, has a greater potential for both research and clinical practice. We appreciate your suggestion and the opportunity to elaborate on our approach.

Comment 4: Statistical modelling concerns: The use of univariate and multivariate regression is questionable given the small number of events relative to the number of covariates. As a rule of thumb, at least 10 events per variable are required to avoid model overfitting. The current stepwise approach risks instability of estimates. I recommend limiting the model to clinically relevant and statistically significant univariate predictors, and clearly justifying their inclusion in the multivariate analysis while preserving all variables to provide a more robust model without overfitting.

Response: Thanks for the comment. We did not use conventional Cox regression, but used Firth’s Cox regression, specifically because of the mentioned rule of thumb. This has been mentioned in the statistical analysis section in the Material and Methods section.

Comment 5: Novelty: Several studies with larger cohorts and direct comparisons to surgical approaches have already been published. The authors should explicitly state what new insights this dataset provides beyond confirming existing evidence. Without a comparator and with the noted methodological constraints, the novelty appears limited.

Response: This is a very important comment. We believe our study has (at least one of) the longest follow-up among similar existing research. At the end, we appreciate the reviewer again for all the comments and helping us improve this draft.

  • End of reviewer 3 comments.

Round 2

Reviewer 1 Report

Comments and Suggestions for Authors

The authors have successfully addressed the Reviewers' comments. No additional concerns at this point.

Reviewer 2 Report

Comments and Suggestions for Authors

The authors had sufficiently taken into consideration this reviewer`s recommendations.